# Systemic complement activation levels in Stargardt disease

**Patty P. A. Dhooge**[1,2], **Esmee H. Runhart**[1,2], **Catherina H. Z. Li**[1,2], **Corrie M. de Kat Angelino**[3], **Carel B. Hoyng**[1], **Renate G. van der Molen**[3], **Anneke I. den Hollander**[1] *

**1** Department of Ophthalmology, Radboud University Medical Center, Nijmegen, The Netherlands,
**2** Donders Institute for Brain, Cognition and Behaviour, Nijmegen, The Netherlands, **3** Department of Laboratory Medicine, Laboratory of Medical Immunology, Radboud University Medical Center, Nijmegen, The Netherlands

* anneke.denhollander@radboudumc.nl

## Abstract

### Purpose

Preclinical research provides evidence for the complement system as a potential common pathway in Stargardt disease (STGD1) and age-related macular degeneration (AMD) leading to retinal pigment epithelium (RPE) loss. However, systemic complement activation has not yet been assessed in STGD1 patients. We conducted a cross-sectional case-control study to assess systemic complement activation in STGD1 patients and its association with disease severity.

### Methods

Systemic concentrations of complement component C3 and its degradation product C3d were compared between 80 STGD1 patients and 80 controls that were frequency matched for age and sex. The C3d/C3 ratio was used as parameter of systemic complement activation. Within the STGD1 cohort, we additionally examined the association between the C3d/C3 ratio, demographic and behavioural factors (age, sex, smoking and BMI), and measures of disease severity (age at onset, visual acuity, and area of atrophy).

### Results

The C3d/C3 ratio did not significantly differ between patients (mean C3d/C3 ratio 3.5±1.4) and controls (mean C3d/C3 ratio 3.6±1.0), mean difference -0.156 (p = 0.804, independent samples t-test). The overall effect size was 8% (95% confidence interval, 3–15%). Elevated C3d/C3 ratios (>8.1) were found in three patients who all had a concomitant inflammatory condition at the time of blood draw. Within the patient cohort, C3 levels were associated with sex (mean difference -134, p = 0.001, independent samples t-test) and BMI (correlation coefficient 0.463, p<0.001, Spearman's Correlation).

### Conclusions

Systemic complement levels were not elevated in STGD1 patients compared to age and sex matched controls and was not associated with STGD1 severity. Considering the

**Data Availability Statement:** The data that support the findings of this study are openly available in the DANS EASY archive at https://doi.org/10.17026/dans-zjn-gvnt (accession number 210703).

**Funding:** Supported by the Foundation Fighting Blindness USA, grant no. PPA-0517-0717-RAD (to F.P.M. Cremers, S. Roosing and C.B.H.). The funders had no role in study design, data collection and analysis, decision to publish, or preparation of the manuscript.

**Competing interests:** The authors have declared that no competing interests exist.

continued absent proof of a systemic contribution of the complement system to RPE loss in STGD1 patients, we hypothesize that complement activation in STGD1 is more likely a local process. In light of upcoming complement-targeted therapies, further studies are needed that measure complement levels in the eye of STGD1 patients.

## Introduction

Stargardt disease (STGD1) is a Mendelian inherited disease in which vision of both eyes progressively deteriorates, a process that is irreversible to date [1,2]. STGD1 is caused by mutations in the adenosine triphosphate binding cassette transporter A4 (*ABCA4)* gene. This gene encodes a transmembrane protein which eliminates retinal from photoreceptor disks and endolysosomes in the retinal pigment epithelium (RPE) [3,4]. In the absence of a functional ABCA4 protein, metabolic byproducts of the visual cycle react to form bisretinoids, the most important of which is N-retinylidene-N-retinylethanolamine (A2E). A2E accumulates in the RPE as a component of lipofuscin, ultimately leading to irreversible degeneration of both RPE and photoreceptors [5–7].

Clinically, there is a substantial heterogeneity in STGD1 phenotypes, age at onset and disease progression [1,2,8]. Even amongst siblings carrying the same *ABCA4* variants, functional outcomes differ considerably [9]. This leads to the question whether other mechanisms, besides the *ABCA4* genotype, are involved in STGD1 pathogenesis. Recent preclinical studies highlight a role for the complement system in the pathophysiology of STGD1. The complement system is part of the innate immune system and protects the human body from pathogens. In cultured RPE cells, photo-oxidation of A2E was shown to activate the complement system and alter the transcription of genes involved in stress response, apoptosis and immune response [10–14]. Moreover, RPE cells of *abca4-/-* mice showed higher levels of complement activation products and increased inflammatory markers as compared to wildtype mice [13,15,16]. Increased levels of C3d, the degradation product of complement component C3, were detected in serum of *Abca4-/-* mice concomitantly to the increase of systemic C3d levels, indicating a relationship between systemic and local complement activation [16]. When treating *Abca4-/-* mice with a targeted gene therapy that increases expression of a complement negative regulatory protein (complement receptor 1-like protein, Crry) in the eye, bisretinoid accumulation and photoreceptor degeneration were reduced [17]. Equally important, the first direct evidence of complement dysregulation in RPE of STGD1 donor eyes was reported recently by immunohistochemical analysis [18]. Taken together, several preclinical studies suggest that complement dysregulation is a contributing factor to photoreceptor degeneration in STGD1.

In age-related macular degeneration (AMD), a disease with similar features as STGD1, the role of the complement system in the pathophysiology is well-established [19]. In contrast to STGD1, AMD is a multifactorial disease including both genetic and behavioural risk factors [20]. The genetic risk for AMD is co-determined by variants in genes encoding components of the complement system (*C3*, *C9*, *CFB*, *CFH*, and *CFI*) [21,22]. In AMD patients, activation of the complement system was not only detected within the eye [23,24], but also systemically in serum and plasma [15,25,26]. The processing of complement factor 3 (C3), measured by the ratio of C3 and its degradation product C3d (C3d/C3), has repeatedly been associated with AMD [27]. Moreover, C3d/C3 levels have been correlated to AMD disease stage [26].

In contrast to extensive research in AMD patients, there is little published data on the contribution of complement dysregulation to the pathogenesis of STGD1. Nevertheless, the results

of the described preclinical studies have already lead to the initiation of a phase 2b clinical trial with a pharmacological agent that targets the complement system by an intravitreal injection with a complement factor C5 inhibitor (avacincaptad pegol/Zimura, NCT03364153). Further understanding of the role of the complement system in STGD1 may shed light on whether complement activation is a local disease process or whether it extends to the systemic circulation, and may contribute to the design of therapeutic approaches for STGD1. Therefore, the current study aimed to assess systemic activation of the complement system in STGD1 patients and investigates whether complement levels are correlated with STGD1 severity.

## Methods

### Participants

For this cross-sectional case-control study, 80 consecutive STGD1 patients and 80 controls without retinal pathology were recruited. Their demographics, blood samples and medical imaging were collected in the Radboud Biobank [28]. In a recent study performed by our group, the C3d/C3 ratio increased with 42% (95% confidence interval (CI) -17% - 103%) when comparing AMD patients with GA to controls. This effect size of 42% was derived from the log C3d/C3 ratio of 1.40 in controls versus 1.57 in AMD patients with GA and a population SD of 0.41 [26]. A sample size of 80 cases and 80 controls is appropriate to reach a statistical power of 80% at a significance level of 0.05, in case of a similar difference as was previously reported in AMD. Simulations for power in relation to patient numbers are provided in S1 Fig. With a cohort of 80 patients and 80 controls, the current study is not sufficiently powered to detect small effect sizes (<20%). Patients were included if they had a STGD1 phenotype defined as bilateral macular dystrophy with presence of yellow-white irregularly shaped flecks corresponding to hyperautofluorescent lesions on short wavelength autofluorescence (SW-AF) and at least one pathogenic *ABCA4* variant.

The control cohort was actively recruited to match both age and sex of the STGD1 cohort. Controls were sampled to match the age and sex frequency of the cases. The control cohort consisted of employees of the Radboud University Medical Center and unrelated spouses of patients within in the eye clinic. To determine that control participants were indeed without retinal pathology, they underwent a full ophthalmic examination including either a fundus photo or spectral domain optical coherence tomography scan.

All participants provided written informed consent before inclusion in the study. Ethical approval was obtained by the CMO Arnhem-Nijmegen and the Radboud University Medical Center local institutional review board (project number: 2017–3535) and this study was conducted according to the tenets of the Declaration of Helsinki. The STROBE guidelines were used to ensure the reporting of this observational study [29].

### Sample collection

Samples were collected according to the standard operating procedures (SOPs) of the Radboud Biobank [28]. Ethylenediaminetetraacetic acid (EDTA) tubes were centrifuged (10 minutes at 2000g at 4˚C), immediately after blood draw. Serum tubes were first placed at room temperature for 1 hour to coagulate and then centrifuged (10 minutes at 2000g at 4˚C). After centrifugation, the obtained plasma and serum samples were kept on dry ice until they were stored in a -80˚C freezer within 2 hours after collection.

### Complement measurements

The level of C3d was quantified in EDTA plasma using an ELISA, as previously described in detail with minor changes [30]. To determine C3d values, pooled activated EDTA plasma of

healthy blood donors was used and set to 100%. This value was calibrated to mg/l using purified C3d containing 10 μg/ml protein. The reference value was determined in a group of 20 healthy volunteers (mean age ± SD; 45.8 ± 11.4) and set at <8.3 mg/l. The C3 concentration was determined in serum by turbidometry (Cobas 8000 platform, F. Hoffman-La Roche Ltd, Basel, Switserland). Reference values for C3 are 700–1500 mg/l. As the initial C3 concentration may influence the C3d level and because of intra-individual variations in the concentration of C3, we used the C3d/C3 ratio as the parameter for complement activation [31]. The reference value for the ratio C3d/C3 is <8.1. The investigators performing the C3 and C3d quantification were masked to patient information.

## Clinical data

Clinical data including demographics (age, sex), use of medication and medical history including information about allergies, previous and current illnesses, surgeries, and abnormal results of physical exams and tests were obtained for all participants. Since inflammatory conditions result in increased complement activation, health complaints and fever in the week preceding blood draw were documented, as well as a history of any chronic inflammatory disease.

To evaluate the factors that might influence activation of the complement system within our STGD1 cohort, we provided the STGD1 patients with a previously used questionnaire [32]. BMI and smoking status (never/past/current and packyears) were documented, because these are known to influence activation of the complement system [15,33].

Measures of disease progression were collected in order to investigate its potential association with systemic complement activation. On the day of the blood draw, visual acuity was measured after refraction using Snellen charts, and a fundus autofluorescence (FAF) image was taken using the Heidelberg Spectralis HRA-OCT (Heidelberg Engineering, Heidelberg, Germany). Moreover, patient records and imaging databases were reviewed to extract the earliest VA measurement and FAF image. To assess the area of RPE atrophy on the FAF images, areas with definitely decreased autofluorescence (DDAF), as defined by Kuehlewein et al. [34], were analysed using the semiautomated RegionFinder software (Heidelberg Engineering, version 2.6.3) according to the method previously described by Schmitz-Valckenberg et al [35]. We used a simplified model of disease progression by considering VA decline and DDAF growth linearly: the difference between the earliest retrospective measurement and the study measurement on the day of blood collection was divided by the number of years between those two measurements.

## Statistical analysis

Statistical analysis was performed using the SPSS statistics package for Windows; version 22 (SPSS IBM, New York, USA). First, C3 levels, C3d levels, the C3d/C3 ratio, age and gender were compared between patients and controls using an independent samples t-test or a chi-square test.

Because of the normal distribution of all complement data in both cases and controls, parametric independent t-tests were used when comparing the complement levels between cases and controls.

Second, Spearman's correlation, ANOVA and independent samples t-test were used to examine associations between the complement levels (C3, C3d and C3d/C3 ratio) and variables related to disease progression (age at onset, VA, DDAF area, VA decline, DDAF growth) as well as demographic and behavioural variables (age, sex, smoking, BMI). The more general non-parametric Spearman's correlation was used to test for associations in order to take the

**Table 1. Complement activation levels in Stargardt disease patients and controls matched for age and sex.**

| Variable | STGD1 (n = 80) | Controls (n = 80) | Mean Difference (95% CI) | P Value |
|---|---|---|---|---|
| Age in years, mean (SD) | 46 (18) | 45 (18) | 0.712 (-4.948–6.373) | 0.804[a] |
| Female sex, n (%) | 44 (55%) | 44 (55%) | not applicable | 1.00[b] |
| C3d in mg/l, mean (SD) | 3.7 (1.3) | 3.7 (0.9) | -0.033 (-0.373–0.308) | 0.851[a] |
| C3 in mg/l, mean (SD) | 1099 (196) | 1057 (155) | 42.5 (-12.6–97,5) | 0.130[a] |
| C3d/C3 ratio, mean (SD) | 3.5 (1.4) | 3.6 (1.0) | -0.156 (-0.537–0.225) | 0.419[a] |

[a]Independent samples t-test

[b]chi-square test. C3, complement component C3; C3d, complement degradation product C3d; CI, confidence interval; SD, standard deviation; STGD1, Stargardt disease.

possibility of a non-linear relationship into account. A p-value of <0.05 was considered significant in this exploratory study.

## Results

The C3 and C3d levels, and the C3d/C3 ratio of 80 patients, 44 females and 36 males, with a mean age of 46±18 years (range 17–84), were compared to 80 age- and sex-matched controls. There was indeed no difference in age (p = 0.804, t-test) or sex (p = 1.00, chi-square test) between both groups. The mean levels of C3d and C3, and the C3d/C3 ratio were similar in both groups (Table 1 and Fig 1).

Six patients (7.5%) had complement activation levels outside the reference window. Four of these patients had abnormal C3 levels (C3 > 1500mg/l) and two patients had abnormal C3d levels (C3d > 8.3mg/l). Both patients with a high C3d level also had a high C3d/C3 ratio. In addition, there was one patient with an elevated C3d/C3 ratio, with normal C3d and C3 levels. Only one individual in the control group (1.3%) had a marginally abnormal complement

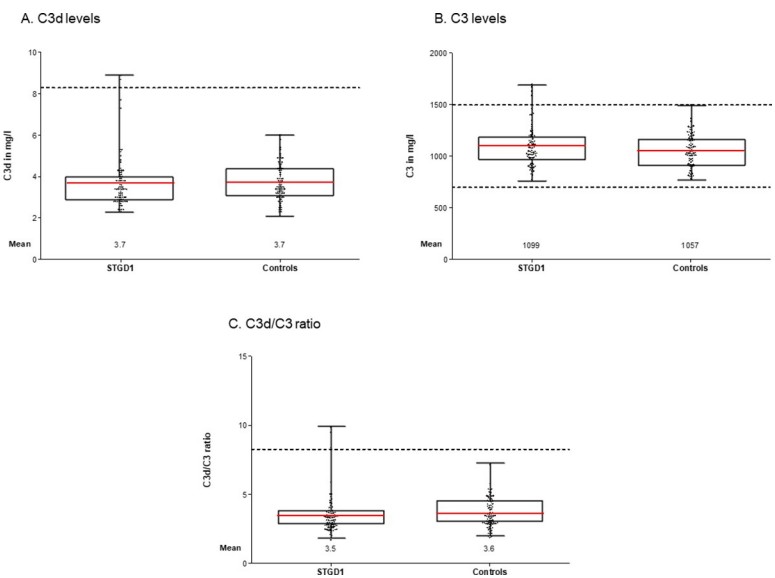

**Fig 1. Complement activation levels in Stargardt disease (STGD1) patients and controls matched for age and sex.**
Level of (A) C3d, (B) C3, (C) C3d/C3 ratio in STGD1 patients and controls. *Dotted lines* represent the reference values. *Red lines* represents the mean. There were no significant differences in means between patients and controls as tested with an independent samples t-test. C3, complement component 3; C3d, complement degradation product C3d.

activation level of C3, namely a C3 of 1501mg/l (reference < 1500mg/l). None of the controls had an abnormal C3d/C3 ratio. With the C3d/C3 ratio as a parameter for complement activation, these findings point towards systemic activation of the complement system in three STGD1 patients and none of the controls.

Since systemic activation of the complement system can be caused by inflammatory conditions, we evaluated the questionnaire data in the participants of the study and specifically in the individuals with altered complement activity. Neither patients, nor controls were on any immunosuppressive medication. Overall, none of the participants reported having had fever in the week before the blood draw. There were 2 patients and 2 controls who reported flu-like symptoms (sore throat, coughing, or running nose). In addition, one STGD1 patient was treated for a urinary tract infection and another STGD1 patient was being treated for breast cancer in the week before the blood draw. Indeed, ten patients and only three controls had a medical history of a chronic inflammatory disease. These conditions included a rheumatic disorder (5 patients and 1 control), bronchitis (4 patients and 2 controls), and ulcerative colitis (1 patient). Of the five STGD1 patients who suffered from a rheumatic disorder, two had active arthritis during blood draw and both of these patients had an elevated C3d/C3 ratio. The third patient with a high C3d/C3 level was treated for a urinary tract infection at the moment of blood draw for this study. To conclude, all three STGD1 patients with an elevated C3d/C3 ratio had a concurrent inflammatory condition during the blood draw.

We next evaluated whether complement activation levels were associated with disease progression and with demographic and behavioral variables. Within the STGD1 cohort, complement levels were not associated with age at onset, VA, DDAF area, VA decline, or DDAF growth as measures of disease progression. The correlation coefficients and corresponding p-values are provided in Table 2 and correlation plots are provided in S2 Fig. C3 levels were associated with sex and BMI in STGD1 patients. Female patients had a higher C3 ratio compared to males (p = 0.001, independent samples t-test). Patients with higher BMI, had higher C3 levels (p<0.001, Spearman's Correlation). Age, smoking status and the number of packyears were not associated with complement activation levels.

## Discussion

Previous preclinical studies indicate a role for the complement system in the pathophysiology of STGD1 [10–17,36,37]. The results of this study did not detect a significant increase or decrease in systemic C3d/C3 ratio in STGD1 patients. Moreover, systemic complement levels were not associated with disease parameters or the rate of disease progression. In 3 of the in total 80 STGD1 patients that did show an increased C3d/C3 ratio a concurrent inflammatory condition most likely explains complement consumption. As we are dealing with a rare disease, we did not exclude participants with symptoms suggestive of underlying inflammation on beforehand in order to give a complete overview of our STGD1 cohort. However, this can be a limitation as it might complicate the understanding of the relationship between complement activation and Stargardt disease. When omitting the participants with flu-like symptoms, medial complaints during blood draw and a medical history of chronic inflammatory disease (thus excluding 12 patients and 5 controls) the mean C3d/C3 ratio was 3.28±0.96 in cases and 3.57±1.06 in controls. Independent samples t-test does not show a significant difference between these means (p = 0.077), which confirms that systemic complement levels are indeed not elevated in STGD1 patients. Therefore, the current study does not provide support for systemic complement activation in the pathophysiology of STGD1.

Previous studies in AMD did find elevated complement levels in patients compared to controls [15,25,26,38]. Moreover, complement levels were associated with disease stage, where

**Table 2. Correlation analysis of all parameters within Stargardt disease patients.**

| Parameter | STGD1 patients (n = 80), Mean (SD) or n (%) | | C3d (mg/l) | C3 (mg/l) | C3d/C3 ratio |
|---|---|---|---|---|---|
| **Cross-sectional parameters related to STGD1 pathology** | | | | | |
| Age at onset in years | 33 (19) | Spearman's Correlation<br>P value | -0.127<br>0.262 | -0.044<br>0.700 | -0.117<br>0.303 |
| Disease duration in years | 13 (12) | Spearman's Correlation<br>P value | -0.061<br>0.589 | 0.020<br>0.863 | -0.030<br>0.789 |
| VA in decimals, average of both eyes | 0.40 (0.40) | Spearman's Correlation<br>P value | -0.130<br>0.249 | -0.020<br>0.858 | -0.127<br>0.261 |
| DDAF area in mm$^2$, average of both eyes | 10 (18) | Spearman's Correlation<br>P value | -0.009<br>0.938 | -0.011<br>0.929 | 0.019<br>0.873 |
| **Retrospective parameters related to STGD1 progression (available for 50 patients)** | | | | | |
| VA decline in decimals/year, average of both eyes | 0.02 (0.08) | Spearman's Correlation<br>P value | 0.138<br>0.311 | -0.010<br>0.941 | 0.134<br>0.324 |
| DDAF growth in mm$^2$/year, average of both eyes | 0.94 (1.53) | Spearman's Correlation<br>P value | -0.014<br>0.926 | -0.230<br>0.108 | 0.094<br>0.517 |
| **Demographic and behavioural factors** | | | | | |
| Age in years | 46 (18) | Spearman's Correlation<br>P value | -0.168<br>0.135 | -0.037<br>0.744 | -0.150<br>0.183 |
| Sex | Males, 36 (45%)<br>Females, 44 (55%) | Level in males, mean (SD)<br>Level in females, mean (SD)<br>Mean Difference (95% CI)<br>T-test, P value | 3.7 (1.1)<br>3.7 (1.4)<br>-0.712 (-0.634–0.492)<br>0.110 | 1025 (112)<br>1159 (228)<br>-134 (-212 - -56)<br>0.001 [a] | 3.6 (1.2)<br>3.3 (1.2)<br>0.302 (-0.315–0.920)<br>0.396 |
| BMI | 25 (4) | Spearman's Correlation<br>P value | 0.124<br>0.273 | 0.463<br><0.001[a] | -0.173<br>0.124 |
| Smoking 3 categories | Current smokers, 11 (14%)<br>Past smokers, 22 (27%)<br>Non-smokers, 47 (59%) | Level in current smokers, mean (SD)<br>Level in past smokers, mean (SD)<br>Level in non-smokers, mean (SD)<br>ANOVA, p value | 4.2 (2.0)<br>3.4 (0.8)<br>3.7 (1.2)<br>0.273 | 1122 (232)<br>1104 (193)<br>1091 (192)<br>0.894 | 4.0 (2.6)<br>3.1 (0.6)<br>3.5 (1.2)<br>0.211 |
| Smoking in packyears | 5 (11) | Spearman's Correlation<br>P value | 0.021<br>0.855 | 0.033<br>0.774 | -0.054<br>0.631 |

[a]p value <0.05. C3, complement component 3; C3d, complement component 3d; CI, confidence interval; SD, standard deviation; STGD1, Stargardt disease.

systemic complement levels increased with consecutive AMD disease stages [26]. AMD patients with geographic atrophy tended to have relatively higher complement activation levels [15,26,38]. However, in the current study complement levels were not elevated in STGD1 patients, who also present with atrophy of the RPE. Yet, previous preclinical studies do provide evidence for a common role for the complement system, leading to RPE loss in both STGD1 and AMD [13,39]. In both diseases, the accumulation of vitamin A byproducts in RPE cells initiates a vicious cycle where chronic low-grade inflammation leads to increased complement activation which, in turn, leads to increased damage to the RPE [13,17,26,36,37]. Once RPE damage occurs, complement factors are exchanged between the blood and retina which contributes to worsening of the disease [16,38]. It is possible that this last step is more likely to occur in AMD patients as compared to STGD1 patients due to contribution of systemic complement levels, which are largely driven by genetic variants in the complement system. In addition, it was previously hypothesized that RPE atrophy acts as a local stimulus that further

amplifies the complement response in AMD [40]. However, we did not find an association between systemic complement levels in STGD1 and RPE atrophy.

Based on mean C3d/C3 ratios in STGD1 cases and controls, and the population SD of 1.21, the current study had an effect size of 8% (95% CI, -5% - 70%). In order to be able to demonstrate a difference in a subsequent study, 2325 participants are needed per cohort. The current study had sufficient power to detect effect sizes that were previously reported in AMD patients (effect size of 42%) [26]. We can thus infer that the effect size of C3d/C3 ratio in STGD1 patients is lower in comparison to AMD patients, or that an effect of C3d/C3 is absent in STGD1. This is likely explained by the difference in genetic predisposition between AMD and STGD1 patients. Genetic variants in the complement genes, most importantly the *CFH* gene, are associated with both increased AMD risk and systemic complement activation [26,40,41]. We did not analyze genetic variants in the complement system in STGD1 patients in the current study, but previous studies did not find an association between the *CFH* genotype and STGD1 and controls [42]. Since we do not detect systemic elevation of C3d/C3 in STGD1 patients, genetic variants in the complement system likely do not play a major role in STGD1, or only contribute to disease in a selected group of STGD1 patients.

To date, very little is known about the relationship between systemic complement activation and local complement activity in the eye. Serum contains the vast majority of complement factors, which are produced by the liver. However, the retina synthesizes several complement proteins locally [43]. Increased complement activity detected in STGD1 in preclinical studies can thus be of local origin, and therefore not measurable in serum. Recently, investigators examined complement activation in the context of RPE from donor eyes clinically diagnosed with STGD1 [18]. However, only one eye was genetically classified as STGD1. For the second donor eye, extracted DNA was of insufficient quality. The third donor eye harbors a *PRPH2* mutation without a *ABCA4* mutation and should therefore be diagnosed as pseudo-Stargardt instead of STGD1 [44–46]. Further studies on intra-ocular complement levels are needed to shed further light on the role of complement in the retina.

In AMD patients, studies on aqueous and vitreous humour have established elevated complement components *in vivo* within the eyes of patients [24,47]. Such studies have not been carried out in STGD1 patients yet, therefore, collecting aqueous and vitreous humor of STGD1 patients would be relevant for future studies. Nevertheless, sample collection is expected to be challenging since only very few STGD1 patients undergo vitrectomy. Alternatively, the ongoing phase 2b clinical trial with complement C5 inhibitor Avacincaptad Pegol (Zimura) [NCT03364153] might prove that local complement dysregulation is a causal factor in the pathogenesis of STGD1, if this therapy successfully rescues the phenotype.

Within the 80 STGD1 patients in this study, no association was detected between systemic complement levels and VA, DDAF area or age at onset. Therefore, systemic complement activation does not explain the clinical heterogeneity within our STGD1 cohort. Prospective studies in which complement levels are measured at several time points during the disease course are necessary to establish the effect of complement levels on disease progression prospectively.

Our study has several limitations. First, we present a cohort that was sufficiently sized to detect effect sizes as large as those previously detected in AMD patients with GA. However, this cohort is too small to detect lower effect sizes. Larger studies are needed to exclude the role of systemic complement activation in STGD1. Second, the inclusion of patients with a possible underlying inflammatory condition creates outliers which causes an extreme data bias. However, when omitting these patients, our results remained the same. Third, complement levels are known to be influenced by other factors including age, sex, BMI and smoking [15,33]. Even though cases and controls were matched on the basis of age and sex, there may be residual confounding and unfortunately, we did not collect BMI and smoking variables for

our control cohort. Therefore, it is not possible to control for these covariates when analyzing our data. This study is the first to investigate systemic complement activation in STGD1 and its potential association with disease progression, and should therefore be considered as a pilot study. Ideally, complement activity should be monitored in individual patients as their disease progresses to establish the effect of the covariates on complement levels and to establish the effect of complement levels on disease progression.

In conclusion, this is the first study to analyze systemic complement activation in STGD1 patients. We did not find evidence that systemic complement activation is associated with STGD1. If the complement system contributes to STGD1 pathogenesis, it might only be locally involved as a response to local processes, or it might be involved on systemic level only in a selected patient group. Since complement inhibitors are being evaluated as a potential therapeutic strategy for STGD1, a better understanding of the role of the complement system in STGD1 patients, by measuring the complement levels within the eye and determining their effect on disease progression, needs to be established. Once we understand the role of the complement system in STGD1 pathogenesis, we can optimize patient selection and intervention timing to maximize the therapeutic effect of complement inhibitors.

## Supporting information

**S1 Fig. Simulations for power in relation to patient numbers.** Panels display different prevalences of an above normal C3d/C3 ratio ($>8.1$) in controls; (A) 1%, (B) 3%, (C) 5%, (D) 10%. Graphs display the number of cases in relation to the statistical power of the study, assuming the 1:1 matched design of the current study. Each graph contains 4 lines with different effect sizes (5%, 10%, 15%, 20%). Very large cohorts are needed to detect small effect sizes.
(TIF)

**S2 Fig. Correlation plots of all parameters within Stargardt disease patients.** Correlation plots of respectively C3d, C3 and C3d/C3 ratio with (A) cross-sectional parameters related to STGD1 pathology, (B) retrospective parameters related to STGD1 progression and (C) demographic and behavioral factors.
(TIF)

## Acknowledgments

The authors would like to thank Johannes Groenewoud for assisting in statistical analysis.

## Author Contributions

**Conceptualization:** Patty P. A. Dhooge, Esmee H. Runhart, Carel B. Hoyng, Renate G. van der Molen, Anneke I. den Hollander.

**Data curation:** Patty P. A. Dhooge, Esmee H. Runhart, Catherina H. Z. Li, Corrie M. de Kat Angelino, Renate G. van der Molen.

**Formal analysis:** Patty P. A. Dhooge, Esmee H. Runhart, Catherina H. Z. Li, Corrie M. de Kat Angelino, Carel B. Hoyng, Renate G. van der Molen, Anneke I. den Hollander.

**Funding acquisition:** Carel B. Hoyng, Renate G. van der Molen.

**Investigation:** Patty P. A. Dhooge, Esmee H. Runhart, Catherina H. Z. Li, Corrie M. de Kat Angelino, Carel B. Hoyng, Renate G. van der Molen, Anneke I. den Hollander.

**Methodology:** Patty P. A. Dhooge, Esmee H. Runhart, Catherina H. Z. Li, Corrie M. de Kat Angelino, Carel B. Hoyng, Renate G. van der Molen, Anneke I. den Hollander.

**Project administration:** Patty P. A. Dhooge, Esmee H. Runhart, Catherina H. Z. Li, Carel B. Hoyng, Renate G. van der Molen, Anneke I. den Hollander.

**Resources:** Patty P. A. Dhooge, Esmee H. Runhart, Carel B. Hoyng, Anneke I. den Hollander.

**Software:** Patty P. A. Dhooge, Esmee H. Runhart, Catherina H. Z. Li, Carel B. Hoyng, Anneke I. den Hollander.

**Supervision:** Carel B. Hoyng, Renate G. van der Molen, Anneke I. den Hollander.

**Validation:** Patty P. A. Dhooge, Esmee H. Runhart, Carel B. Hoyng, Anneke I. den Hollander.

**Visualization:** Patty P. A. Dhooge, Esmee H. Runhart, Carel B. Hoyng, Anneke I. den Hollander.

**Writing – original draft:** Patty P. A. Dhooge, Esmee H. Runhart, Anneke I. den Hollander.

**Writing – review & editing:** Patty P. A. Dhooge, Esmee H. Runhart, Catherina H. Z. Li, Corrie M. de Kat Angelino, Carel B. Hoyng, Renate G. van der Molen, Anneke I. den Hollander.

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
