## [Decision Letter · Decision Letter 0]

13 Apr 2021

PONE-D-21-08972

Systemic complement activation levels in Stargardt disease

PLOS ONE

Dear Dr. den Hollander,

Thank you for submitting your manuscript to PLOS ONE. After careful consideration, we feel that it has merit but does not fully meet PLOS ONE’s publication criteria as it currently stands. Therefore, we invite you to submit a revised version of the manuscript that addresses the points raised during the review process.

Two reviewers found this a very interesting study and made suggestions to improve it. We look forward to the revised manuscript !

We look forward to receiving your revised manuscript.

Kind regards,

Demetrios G. Vavvas

Academic Editor

PLOS ONE

Journal Requirements:

2. Thank you for including your ethics statement:  "Ethical approval was obtained by the local institutional review board (project number: 2017-3535). All participants provided written informed consent before inclusion in the study. ".   

Reviewers' comments:

Reviewer's Responses to Questions

**Comments to the Author**

1. Is the manuscript technically sound, and do the data support the conclusions?

Reviewer #1: Yes

Reviewer #2: Partly

2. Has the statistical analysis been performed appropriately and rigorously? 

Reviewer #1: No

Reviewer #2: Yes

3. Have the authors made all data underlying the findings in their manuscript fully available?

Reviewer #1: Yes

Reviewer #2: No

4. Is the manuscript presented in an intelligible fashion and written in standard English?

Reviewer #1: Yes

Reviewer #2: Yes

5. Review Comments to the Author

Reviewer #1: In this manuscript, Dhooge et al. evaluate the systemic complement activation profiles of patients with Stargardt disease compared to controls without retinal pathology. This topic is clinically of interest since there are preclinical studies, predominately in animal models, that suggest that complement dysregulation may play a role in the pathogenesis of Stargardt disease. Furthermore, there is a human study, as the authors cite, investigating the use of the C5 inhibitor avacincaptad pegol in patients with Stargardt disease. As such, this is a timely study that has the potential to enhance our understanding of the role that complement dysregulation may play in Stargardt disease and to inform the development of future clinical trials. The authors are encouraged to address the following concerns.

Major concerns

1. Line 90: Selection of controls is a very important consideration for any case-control study. Can the authors provide further details regarding how these control participants were recruited? The manuscript mentions that they are “without retinal pathology.” How was this determined? To what extent did the authors investigate whether the control patients had any other medical/ophthalmic co-morbidities (e.g., self-report, ophthalmic examination, or otherwise)? If they were clinic-/hospital-based controls, the authors should consider the possibility of selection bias (Berkson paradox) and how this may have influenced their findings. Further details by the authors would enable a reader to assess the possibility of bias; limitations should also be detailed further in the Discussion.

2. Lines 117-119, Lines 168-172: The authors acknowledge the important point that other factors (e.g., concurrent infection, autoimmune disease) can influence systemic complement activation profiles. This reviewer is curious why the authors did not exclude participants with symptoms suggestive of underlying infection or chronic autoimmune disease, as inclusion of these participants may complicate understanding of the relationship between complement activation and Stargardt disease. Is it known whether cases or controls were on any immunosuppressive medications that could alter systemic complement activation? This is a major limitation of the present study that complicates interpretation of the central findings. Would it be possible for the authors to perform a sensitivity analysis of omitting the participants with such features to determine whether their results remain the same? This should also be further elucidated in the Discussion.

3. As the authors mention, complement activation is heavily influenced by other factors, including age, sex, BMI, and smoking. Despite the fact that the controls were age- and sex-matched to cases, there may be residual confounding that still warrants controlling for (https://www.bmj.com/content/352/bmj.i969), especially since there were differences based on sex in the cases (Line 187-188). The fact that there was no statistically significant difference between age or sex between cases and controls does not rule out the possibility that they may be influencing the results. As such, the authors are encouraged to repeat their analysis but control for the covariates of age, sex, BMI, and smoking, for example, with general linear models. This is not likely to change the ultimate conclusions given that the authors do not find a significant difference but is still worth considering for the sake of completeness and rigorous statistical methodology.

Minor concerns:

Line 83: Given that the patients were not followed longitudinally, I think it would be more appropriate to characterize this study as a cross-sectional study here and elsewhere.

Line 105: Were the investigators performing C3/C3d quantitation masked to patient information?

Line 128-135: Could the authors provide more specific information of what is meant by “first available” VA/FAF image? Does this mean the most recent or the earliest? Was the interval similar from patient to patient? This may affect the accuracy of the authors’ simplified model of assuming linear decline.

Line 138-144: Throughout the manuscript, could the authors provide more specific rationale for their selection of statistical testing. In some cases, they have used parametric independent t-tests but, elsewhere, they have used the non-parametric Spearman correlation test to evaluate similar data. Fisher’s exact test is often used instead of the chi-square test for small samples sizes; was there a specific reason that this was chosen?

Table 2: Please show the correlation plots of the relationships described in Table 2, which would enhance the ability of the reader to visualize the relationships or lack thereof.

Reviewer #2: # Overview

I would like to thank the authors for working on a very interesting question with particularly exciting therapeutic implications. Not being a subject expert, I can only comment on the epidemiological and methodological merits of this study.

This study uses a case-control design of 80 confirmed Stargardt cases and 80 age and sex matched controls to identify no clinically or statistically significant differences in C3d/C3 between the two.

This manuscript correctly uses a case-control design to study a rare disease and commendably presents power calculations. However, it also suffers from (a) the absence of more elaborate power analyses necessitated by the null findings of this study, (b) the lack of clarity in how controls were identified and sampled and (c) the lack of robust associational analyses between multiple potential risk factors and C3d/C3.

Overall, the epidemiological/methodological aspects of this manuscript require a major revision before it can meaningfully contribute to the biomedical literature.

# Major concerns

1. **Power analysis.** Methods, Lines 85-87. The authors should be congratulated for including a power analysis. This power analysis is extraordinarily important in this article because it did not identify a statistically significant difference between cases and controls. As such, we need a few more details. First, we need to know what effect size this power refers to - merely stating that this is a "similar difference as was previously reported in AMD" is inadequate. Some numbers are mentioned in the Discussion (line 220), but we need the effect size, not the numbers (e.g. an increase of 20% in C3d/C3 ratio between controls and cases). Second, what was the assumed significance level? I presume this was 0.05, but it has to be clarified. Third, please provide a figure that consists of 4 panels of graphs. Each graph should represent a different prevalence of "high" C3d/C3 ratio in the controls to cover 4 realistic scenarios (e.g. 1%, 3%, 5%, 10% prevalence). The horizontal axis of each graph should be the number of cases, the vertical axis should be the power and each graph should contain 4 lines referring to 4 different realistic effect sizes (e.g. 5% vs. 10% vs. 15% vs. 20% increase). These graphs would assume the 1:1 matched design of the current study. Such a figure would help us understand how small an effect size this study was powered to detect and what we would need to do if we wanted to power a study for an even smaller effect size (which seems to be the case in this study).

2. **Lack of control description.** Methods, Lines 90-91. (a) It is unclear whether this refers to individual matching (e.g. a 40 year old male STGD1 patient was matched to a 40 year old male control) vs. frequency matched (e.g. the controls were sampled to match the age and sex frequency of the cases). This matters because the two strategies necessitate different analytic approaches. I have assessed this manuscript assuming that frequency matching was used. Please clarify the type of matching here and in the abstract. (b) We need a lot more information about how controls were chosen. After recall bias, the second most important thread to the validity of a case-control study is the choice of controls. Where were these controls found? How were they sampled? What were the eligibility criteria?

3. **Poorly interpretable associational analysis.** Results, Lines 183-190. These results are very difficult to interpret using such analyses. If the authors wish to retain this work, I recommend that they replace it with a univariable regression analysis (i.e. use a regression to identify the associational effect of each variable on C3, C3d and their ratio) and supplement it with a multivariable regression analysis. Both analyses should also include a feature for whether each individual is a case or control to respect the design of this study. The authors can also consider excluding this analysis altogether as it does not contribute to the scientific importance of this work and its main hypothesis of interest.

4. **Need to use the STROBE reporting guidelines.** It is extremely important that the authors use the STROBE reporting guidelines (https://www.equator-network.org/wp-content/uploads/2015/10/STROBE_checklist_v4_combined.pdf) to make sure that they are reporting on all aspects of their study. As can be seen throughout my comments, there is a number of missing pieces. Please also cite the STROBE guidelines to promote trust in your work.

5. **Need to make all non-PII data and code available.** As per PLOS One guidelines, please make all data with non-personally identifiable information available on repositories like Zenodo. Similarly, please make all of your code available on repositories like Zenodo, GitHub or OSF (Open Science Framework).

# Minor concerns.

1. Abstract, Line 16. Please report the overall effect size and its confidence interval, in addition to the means and variance within the two groups.

2. Abstract, Lines 19-20. Please report the mean difference and correlation in addition to the p-values.

3. Methods, Line 83 and Line 122. What does the manuscript mean with "prospective case-control study"? Does it mean that the exposures (i.e. C3d, C3) and risk factors (e.g. BMI, smoking, etc.) were recorded from medical records that predate the development of disease? If yes, then this is indeed a prospective case-control study, otherwise, this is a retrospective case-control study. In Methods Line 122, the use of a questionnaire suggests that this is probably a retrospective case-control study. Note that the reason we distinguish between retrospective and prospective case-control studies is one of the major caveats of such studies, namely "recall bias". Prospective studies use exposures that were recorded before the disease status was known, hence are unlikely to suffer from recall bias, whereas retrospective case-control studies use questionnaires to identify the status of exposures, hence are likely to suffer from recall bias.

4. Methods, Line 116. What medical history was obtained? This does not seem to be mentioned or used anywhere in the manuscript, yet it is important in understanding how similar the cases are to controls.

5. Methods, Line 140. It does not make sense to use a t-test and a Spearman's correlation. Spearman's correlation is used to avoid certain parametric assumptions, which are made by the t-test. As such, either use a non-parametric equivalent of the t-test (e.g. Mann-Whitney U Test) and Spearman's correlation, or the t-test and a parametric equivalent of the Spearman's correlation (e.g. Pearson's correlation). Unless given a good reason, this study should be using the t-test and Pearson's correlation, which are more powerful tests.

6. Results, Line 149. The Fisher's exact test was not mentioned in the statistical analysis. As per my above comment, please stick to parametric tests unless there is a good reason not to. Also, please add another column to this table to report the mean difference between the two groups and its 95% confidence interval - these are the numbers we really care about.

7. Results, Lines 151-152. Please report comparisons for ALL variables collected, not just the age, sex and C3d/C3 (e.g. smoking status, BMI, etc.).

8. Results, Lines 153-157. These dot plots are very helpful, thank you. Please superimpose these dots with a box plot to illustrate also the first and third quartiles.

9. Results, Lines 168-181. This paragraph confirms that the cases may indeed come from a qualitatively different "population' than the controls. As noted above, it is extremely important to describe how cases and controls were sampled, from where, etc.

10. Discussion, Line 196. Please clarify that by "disease progression" this manuscript refers to "rate of progression".

11. Discussion. Please include a limitations section here. Please state at least the three most important limitations and how robust this study may be to these limitations.

6. PLOS authors have the option to publish the peer review history of their article (what does this mean?). If published, this will include your full peer review and any attached files.

Reviewer #1: No

Reviewer #2: **Yes: **Stylianos Serghiou

---

## [Author Response · Author response to Decision Letter 0]

25 May 2021

We would like to thank the editor and the reviewers for their additional comments to improve our manuscript. We uploaded a point-by-point response to their additional comments named Response to Reviewers. Please refer to this file which is easier to read due to layout. We hope you now find our manuscript suitable for publication in PLOS ONE.

Suggestion, Question, or Comment from Reviewer #1 

1. Line 90: Selection of controls is a very important consideration for any case-control study. Can the authors provide further details regarding how these control participants were recruited? The manuscript mentions that they are “without retinal pathology.” How was this determined? To what extent did the authors investigate whether the control patients had any other medical/ophthalmic co-morbidities (e.g., self-report, ophthalmic examination, or otherwise)? If they were clinic-/hospital-based controls, the authors should consider the possibility of selection bias (Berkson paradox) and how this may have influenced their findings. Further details by the authors would enable a reader to assess the possibility of bias; limitations should also be detailed further in the Discussion. 

We provided additional details about the control cohort in the methods part.

Because we did not include clinic-/hospital-based controls, we do not expect a selection bias. We did provide a limitation paragraph in the discussion section, however as we do not expect a selection bias, this was not included in this limitation paragraph.

Methods, participants, line 90, following text was added:

The control cohort was actively recruited to match both age and sex of the STGD1 cohort. Controls were sampled to match the age and sex frequency of the cases. The control cohort consisted of employees of the Radboud University Medical Center and unrelated spouses of patients within in the eye clinic. To determine that control participants were indeed without retinal pathology, they underwent a full ophthalmic examination including either a fundus photo or spectral domain optical coherence tomography scan. 

2. Lines 117-119, Lines 168-172: The authors acknowledge the important point that other factors (e.g., concurrent infection, autoimmune disease) can influence systemic complement activation profiles. This reviewer is curious why the authors did not exclude participants with symptoms suggestive of underlying infection or chronic autoimmune disease, as inclusion of these participants may complicate understanding of the relationship between complement activation and Stargardt disease. Is it known whether cases or controls were on any immunosuppressive medications that could alter systemic complement activation? This is a major limitation of the present study that complicates interpretation of the central findings. Would it be possible for the authors to perform a sensitivity analysis of omitting the participants with such features to determine whether their results remain the same? This should also be further elucidated in the Discussion. 

We did not exclude participants with symptoms suggestive of underlying infection beforehand, because we wanted to give a complete overview of our STGD1 cohort as we are dealing with a rare disease. 

We are well aware that this complicates the understanding of the relationship between complement activation and STGD1. This is why we elaborated on the concomitant diseases with a full paragraph in the results section (lines 168 – 181).

Neither cases, nor controls were on any immunosuppressive medication at time of inclusion. The patient with breast cancer is being treated with anastrozole, a hormone therapy that does not modulate the immune system.

We performed a sensitivity analysis omitting patients with symptoms suggestive of underlying infection (12 patients and 5 controls). The mean C3d/C3 ratio was 3.28±0.96 in cases and 3.57±1.06 in controls. Independent samples t-test did not show a significant difference between these means (p = 0.077). The sensitivity analysis does not change our ultimate conclusions and confirms that systemic complement levels are indeed not elevated in STGD1 patients. 

Results, line 170, text added: Neither patients, nor controls were on any immunosuppressive medication.

Discussion, line 199, text added: As we are dealing with a rare disease, we did not exclude participants with symptoms suggestive of underlying inflammation on beforehand in order to give a complete overview of our STGD1 cohort. However, this can be a limitation as it might complicate the understanding of the relationship between complement activation and Stargardt disease. When omitting the participants with flu-like symptoms, medial complaints during blood draw and a medical history of chronic inflammatory disease (thus excluding 12 patients and 5 controls) the mean C3d/C3 ratio was 3.28±0.96 in cases and 3.57±1.06 in controls. Independent samples t-test does not show a significant difference between these means (p = 0.077), which confirms that systemic complement levels are indeed not elevated in STGD1 patients. 

3. As the authors mention, complement activation is heavily influenced by other factors, including age, sex, BMI, and smoking. Despite the fact that the controls were age- and sex-matched to cases, there may be residual confounding that still warrants controlling for (https://www.bmj.com/content/352/bmj.i969), especially since there were differences based on sex in the cases (Line 187-188). The fact that there was no statistically significant difference between age or sex between cases and controls does not rule out the possibility that they may be influencing the results. As such, the authors are encouraged to repeat their analysis but control for the covariates of age, sex, BMI, and smoking, for example, with general linear models. This is not likely to change the ultimate conclusions given that the authors do not find a significant difference but is still worth considering for the sake of completeness and rigorous statistical methodology. 

We agree that general linear models or regression analysis would complete the statistical methods. However, there are 3 important reasons why we did not include these in the manuscript.

1) With the current data, it is only possible to control for the covariates age and sex, and not for BMI and smoking (when comparing STGD1 to controls), because we did not collect BMI and smoking variables for our control cohort. It is also not possible to do a multivariable regression analysis with a feature indicating if an individual is a case or control. The more is asked from control participants, the less likely they are willing to cooperate. Therefore we asked as little as possible from the control group. At the moment of data collection we were only interested in the additional factors that might influence complement activation within the STGD1 cohort and limited the BMI and smoking question to our STGD1 cohort (together with parameters related to STGD1, which are not present in controls). We now see that this was not clearly written in the manuscript and we have changed the text accordingly. 

Since we cannot correct for all possible covariates, we did not add this calculation to the manuscript because it wouldn't give a complete picture.

2) Repeating the association analysis with general linear models or multivariate regression for the variables in table 2 is difficult, because the exact role of the complement system within STGD1 is still not clear. When using regression or linear models you need to define cause and consequence. However, we are not sure if atrophy leads to higher complement levels, or that, if a person has higher complement levels from other causes (BMI/sex), he/she also develops more atrophy. We hypothesize that increased complement activation leads to increased damage to the RPE. This increased damage will lead to lower visual acuity and larger areas of geographic atrophy. This leads to the following sequence:

Age/Sex/BMI/Smoking � Complement levels � RPE damage � VA & DDAF

Complement level is an intermediary variable in this sequence. Therefore we cannot include all variables in Table 2 in one model. Correcting for age/sex/BMI/smoking when evaluating whether systemic complement levels are associated with disease parameters or disease progression is not possible in this way.

3) Using general linear models or regression analysis, one assumes that relationships are linear. We used Spearman’s correlation to determine more general relationships between values. Since we don’t find any correlation between C3d/C3 ratio and the values in table 2, it is unlikely that we find an association with general linear models or univariate regression. Adding general linear models or univariate regression analysis would therefore not change the ultimate conclusion in this manuscript. We prefer to use Spearman’s correlation to look at more general relationships between values.

This study is the first to investigate the potential association between systemic complement activation and disease progression and can therefore be seen as a pilot study. Prospective studies in which complement levels are measured at several time points during the disease course are necessary to establish the effect of covariates on complement levels and to establish the effect of complement levels on disease progression. This is also stated in the discussion section. 

Methods, Clinical Data, lines 119-122: ‘In addition to … 

through a questionnaire.’ was changed to ‘To evaluate the factors that might influence activation of the complement system within our STGD1 cohort, we provided the STGD1 patients with a previously used questionnaire. [31] BMI and smoking status (never/past/current and packyears) were documented, because these are known to influence activation of the complement system.[15, 32]’ 

Discussion, line 256, limitations paragraph added:

Our study has several limitations … Third, complement levels are known to be influenced by other factors including age, sex, BMI and smoking. [15, 32] Even though cases and controls were matched on the basis of age and sex, there may be residual confounding and unfortunately, we did not collect BMI and smoking variables for our control cohort. Therefore it is not possible to control for these covariates when analyzing our data. This study is the first to investigate systemic complement activation in STGD1 and its potential association with disease progression, and should therefore be considered a pilot study. Ideally, complement activity should be monitored in individual patients as their disease progresses to establish the effect of the covariates on complement levels and to establish the effect of complement levels on disease progression.

4. Line 83: Given that the patients were not followed longitudinally, I think it would be more appropriate to characterize this study as a cross-sectional study here and elsewhere. 

We agree and changed prospective case-control study to cross-sectional case-control study. 

Abstract line 6: ‘We conducted a prospective case-control study’ was changed to: We conducted a cross-sectional case-control study.

Results line 83: For this case-control study was changed to for this cross-sectional case-control study.

5. Line 105: Were the investigators performing C3/C3d quantitation masked to patient information? Yes, these investigators were masked. 

We added this information to the manuscript.

Methods, complement measurements, line 113, text added: The investigators performing the C3 and C3d quantification were masked to patient information.

6. Line 128-135: Could the authors provide more specific information of what is meant by “first available” VA/FAF image? Does this mean the most recent or the earliest? Was the interval similar from patient to patient? This may affect the accuracy of the authors’ simplified model of assuming linear decline.

In this case, ‘first available’ means ‘the earliest’. 

The used interval differs between patients and depends on the age at diagnosis and disease duration. Therefore, the simplified model gives a value for the progression per year.

For atrophy measurements it is known that atrophy radius enlarges at a constant rate of approximately 0.163 mm/year, independent of time.(PMID: 3104722) VA progression is not linear in STGD1 and different intervals may indeed affect the accuracy of the simplified model. However, until now there is no easy way to assess the VA progression using one variable. As we state in the discussion, prospective studies in which complement levels are measured at several time points during the disease course are necessary to establish the effect of complement levels on disease progression prospectively. 

Methods, clinical data, line 128: ‘patient records and imaging databases were reviewed to extract the first available VA measurement and FAF image’ was changed to: patient records and imaging databases were reviewed to extract the earliest VA measurement and FAF image.

Methods, clinical data, line 134: ‘the difference between the first retrospective measurement…’ was changed to: the difference between the earliest retrospective measurement.

7. Line 138-144: Throughout the manuscript, could the authors provide more specific rationale for their selection of statistical testing. In some cases, they have used parametric independent t-tests but, elsewhere, they have used the non-parametric Spearman correlation test to evaluate similar data. Fisher’s exact test is often used instead of the chi-square test for small samples sizes; was there a specific reason that this was chosen? 

Our current data show a normal distribution for all the complement data in both cases and controls (C3, C3d and C3d/C3 ratio). Because of the normal distribution of the data, we can use parametric independent t-tests when comparing the complement levels between cases and controls (Table 1).

For Table 2 we specifically used non-parametric tests. The complement values are normally distributed, but the other tested values (VA,DDAF) are often not normally distributed. Besides, we assume that the relationship between all tested values in Table 2 might not be linear. The parametric Pearson’s correlation only evaluates linear relationships, while the non-parametric Spearman’s correlation is more general. We therefore used Spearman’s correlation to evaluate more general relationships between values. 

Fischer’s exact was only used to compare sex between cases and controls. We agree that chi-square test is a better fit for small samples sizes. We therefore changed the analysis accordingly. When comparing sex between cases and controls with Chi-square instead of Fisher’s exact, p value remains 1.00.

We adjusted the statistical analysis section in methods to clarify our rationale. 

Methods, Statistical Analysis, lines 138-144 was changed to: Statistical analysis was performed using the SPSS statistics package for Windows; version 22 (SPSS IBM, New York, USA). First, C3 levels, C3d levels, the C3d/C3 ratio, age and gender were compared between patients and controls using an independent samples t-test or a chi-square test. Because of the normal distribution of all complement data in both cases and controls, parametric independent t-tests were used when comparing the complement levels between cases and controls. Second, Spearman’s correlation, ANOVA and independent samples t-test were used to examine associations between the complement levels (C3, C3d and C3d/C3 ratio) and variables related to disease progression (age at onset, VA, DDAF area, VA decline, DDAF growth) as well as demographic and behavioural variables (age, sex, smoking, BMI). The more general non-parametric Spearman’s correlation was used to test for associations in order to take the possibility of a non-linear relationship into account. A p-value of <0.05 was considered significant in this exploratory study.

Results, lines 148-149: ‘There was .. both groups’ was changed to There was indeed no difference in age (p=0.804, t-test) or sex (p=1.00, chi-square test) between both groups.

Table 1, line 151: ‘Fischer’s exact’ was replaced by chi-square test. 

8. Table 2: Please show the correlation plots of the relationships described in Table 2, which would enhance the ability of the reader to visualize the relationships or lack thereof. 

We provided correlation plots in supporting Fig S2. 

Supporting Fig S2 was added to the manuscript.

Suggestion, Question, or Comment from Reviewer #2 

1. **Power analysis.** Methods, Lines 85-87. The authors should be congratulated for including a power analysis. This power analysis is extraordinarily important in this article because it did not identify a statistically significant difference between cases and controls. As such, we need a few more details. First, we need to know what effect size this power refers to - merely stating that this is a "similar difference as was previously reported in AMD" is inadequate. Some numbers are mentioned in the Discussion (line 220), but we need the effect size, not the numbers (e.g. an increase of 20% in C3d/C3 ratio between controls and cases). Second, what was the assumed significance level? I presume this was 0.05, but it has to be clarified. Third, please provide a figure that consists of 4 panels of graphs. Each graph should represent a different prevalence of "high" C3d/C3 ratio in the controls to cover 4 realistic scenarios (e.g. 1%, 3%, 5%, 10% prevalence). The horizontal axis of each graph should be the number of cases, the vertical axis should be the power and each graph should contain 4 lines referring to 4 different realistic effect sizes (e.g. 5% vs. 10% vs. 15% vs. 20% increase). These graphs would assume the 1:1 matched design of the current study. Such a figure would help us understand how small an effect size this study was powered to detect and what we would need to do if we wanted to power a study for an even smaller effect size (which seems to be the case in this study). 

The effect size in our recent study performed by our group in AMD was 42%. (PMID: 32176267) The assumed significance level was 0.05.

We clarified the power calculation accordingly. 

We provided the requested figure as supporting Fig S1. 

Methods, Participants, line 85-87: ‘With a sample … in house study’ was changed to In a recent study performed by our group, the C3d/C3 ratio increased with 42% (95% confidence interval (CI) -17% - 103%) when comparing AMD patients with GA to controls. This effect size of 42% was derived from the log C3d/C3 ratio of 1.40 in controls versus 1.57 in AMD patients with GA and a population SD of 0.41.[26] A sample size of 80 cases and 80 controls is appropriate to reach a statistical power of 80% at a significance level of 0.05, in case of a similar difference as was previously reported in AMD. Simulations for power in relation to patient numbers are provided in supporting Fig S1. With a cohort of 80 patients and 80 controls, the current study is not sufficiently powered to detect small effect sizes (<20%).

Supporting Fig S1 was added to the manuscript.

2. **Lack of control description.** Methods, Lines 90-91. (a) It is unclear whether this refers to individual matching (e.g. a 40 year old male STGD1 patient was matched to a 40 year old male control) vs. frequency matched (e.g. the controls were sampled to match the age and sex frequency of the cases). This matters because the two strategies necessitate different analytic approaches. I have assessed this manuscript assuming that frequency matching was used. Please clarify the type of matching here and in the abstract. (b) We need a lot more information about how controls were chosen. After recall bias, the second most important thread to the validity of a case-control study is the choice of controls. Where were these controls found? How were they sampled? What were the eligibility criteria? 

Controls were sampled to match the age and sex frequency of the cases. 

We provided additional details about the control cohort in the Methods section.

Please see our previous answer to reviewer 1 point 1. Abstract, line 10: ‘80 controls matched for age and sex’ was changed to 80 controls that were frequency matched for age and sex.

 3. **Poorly interpretable associational analysis.** Results, Lines 183-190. These results are very difficult to interpret using such analyses. If the authors wish to retain this work, I recommend that they replace it with a univariable regression analysis (i.e. use a regression to identify the associational effect of each variable on C3, C3d and their ratio) and supplement it with a multivariable regression analysis. Both analyses should also include a feature for whether each individual is a case or control to respect the design of this study. The authors can also consider excluding this analysis altogether as it does not contribute to the scientific importance of this work and its main hypothesis of interest. 

Indeed, our main objective is to compare the systemic complement levels between cases and controls. Our second objective is to evaluate whether systemic complement levels are associated with disease parameters or disease progression. This study is the first to investigate this potential association. In our opinion, the correlation analysis in Table 2 is scientifically important and prefer not to exclude this analysis. 

Please see our previous answer to reviewer 1 point 3. 

4. **Need to use the STROBE reporting guidelines.** It is extremely important that the authors use the STROBE reporting guidelines (https://www.equator-network.org/wp-content/uploads/2015/10/STROBE_checklist_v4_combined.pdf) to make sure that they are reporting on all aspects of their study. As can be seen throughout my comments, there is a number of missing pieces. Please also cite the STROBE guidelines to promote trust in your work. 

We have checked our work with the STROBE reporting guidelines and cited the guidelines in the Methods section.

Methods, line 94, sentence added: The STROBE guidelines were used to ensure the reporting of this observational study.[29]

5. **Need to make all non-PII data and code available.** As per PLOS One guidelines, please make all data with non-personally identifiable information available on repositories like Zenodo. Similarly, please make all of your code available on repositories like Zenodo, GitHub or OSF (Open Science Framework). 

We deposited the data in the Radboud Repository. Our data files will first be checked by our institutional Radboud Information Systems Servicedesk. After the check, the dataset will be submitted to the DANS EASY archive and will be publicly available. This process takes some time. We will provide the DOI as soon as possible. 

6. Abstract, Line 16. Please report the overall effect size and its confidence interval, in addition to the means and variance within the two groups. 

We reported these values. Effect size was also added to the discussion section. 

Results, line 16: ‘The C3d/C3 … controls (3.6±1.0)’ was changed to The C3d/C3 ratio did not significantly differ between patients (mean C3d/C3 ratio 3.5±1.4) and controls (mean C3d/C3 ratio 3.6±1.0), mean difference -0.156, p=0.804, independent samples t-test. The overall effect size was 8% (95% confidence interval, 3-15%). 

Discussion, line 219, sentence added: Based on mean C3d/C3 ratios in STGD1 cases and controls, and the population SD of 1.21, the current study had an effect size of 8% (95% CI, -5% - 70%). In order to be able to demonstrate a difference in a subsequent study, 2325 participants are needed per cohort. The current study had sufficient power to detect effect sizes that were previously reported in AMD patients (effect size of 42%).[26]

7. Abstract, Lines 19-20. Please report the mean difference and correlation in addition to the p-values. 

We reported these values. 

Abstract, Results, lines 16-21: ‘The C3d/C3 ratio … smoking status’ was changed to The C3d/C3 ratio did not significantly differ between patients (mean C3d/C3 ratio 3.5±1.4) and controls (mean C3d/C3 ratio 3.6±1.0), mean difference -0.156 (p=0.804, independent samples t-test). The overall effect size was 8% (95% confidence interval, 3-15%). Elevated C3d/C3 ratios (>8.1) were found in three patients who all had a concomitant inflammatory condition at the time of blood draw. Within the patient cohort, C3 levels were associated with sex (mean difference -134, p=0.001, independent samples t-test) and BMI (correlation coefficient 0.463, p<0.001, Spearman’s Correlation).

8. Methods, Line 83 and Line 122. What does the manuscript mean with "prospective case-control study"? Does it mean that the exposures (i.e. C3d, C3) and risk factors (e.g. BMI, smoking, etc.) were recorded from medical records that predate the development of disease? If yes, then this is indeed a prospective case-control study, otherwise, this is a retrospective case-control study. In Methods Line 122, the use of a questionnaire suggests that this is probably a retrospective case-control study. Note that the reason we distinguish between retrospective and prospective case-control studies is one of the major caveats of such studies, namely "recall bias". Prospective studies use exposures that were recorded before the disease status was known, hence are unlikely to suffer from recall bias, whereas retrospective case-control studies use questionnaires to identify the status of exposures, hence are likely to suffer from recall bias. We agree that prospective case-control study is not the correct term. 

Patients were included if STGD1 diagnosis was already established. Blood was drawn to collect C3d and C3 exposures and we used a questionnaire to record the other exposures (BMI, smoking, medication) at the time of blood draw. Imaging was performed at the same day as the blood draw. In conclusion, all variables were collected at one given point in time. The patients were not followed longitudinally. 

We therefore changed the characterization of our study to a cross-sectional case-control study.

Abstract line 6: ‘We conducted a prospective case-control study’ was changed to: We conducted a cross-sectional case-control study.

Results line 83: For this case-control study was changed to for this cross-sectional case-control study.

9. Methods, Line 116. What medical history was obtained? This does not seem to be mentioned or used anywhere in the manuscript, yet it is important in understanding how similar the cases are to controls. 

We obtained a personal medical history including information about allergies, previous and current illnesses, surgeries, and abnormal results of physical exams and tests for both cases and controls. We specified this in the Methods section

Besides, we specifically asked participants about health complaints and fever in the week preceding blood draw as well as a history of any chronic inflammatory disease since inflammatory conditions result in increased complement activation. This was specified in lines 117-119. 

Methods, lines 116-117:‘Clinical data including .. obtained for all patients’ was changed to Clinical data including demographics (age, sex), use of medication and medical history including information about allergies, previous and current illnesses, surgeries ,and abnormal results of physical exams and tests were obtained for all participants.

10. Methods, Line 140. It does not make sense to use a t-test and a Spearman's correlation. Spearman's correlation is used to avoid certain parametric assumptions, which are made by the t-test. As such, either use a non-parametric equivalent of the t-test (e.g. Mann-Whitney U Test) and Spearman's correlation, or the t-test and a parametric equivalent of the Spearman's correlation (e.g. Pearson's correlation). Unless given a good reason, this study should be using the t-test and Pearson's correlation, which are more powerful tests. 

Please see our previous answer to reviewer 1 point 7.

11. Results, Line 149. The Fisher's exact test was not mentioned in the statistical analysis. As per my above comment, please stick to parametric tests unless there is a good reason not to. Also, please add another column to this table to report the mean difference between the two groups and its 95% confidence interval - these are the numbers we really care about. 

We added another column to Table 1 that includes mean difference and the 95% confidence interval. 

Indeed, Fisher’s exact was not mentioned in the statistical analysis. We agreed with reviewer 1 that chi-square test is a better fit for small samples sizes. We therefore changed the analysis accordingly.

We adjusted the statistical analysis section in the methods section and included the chi-square test.

Please see our previous answer to reviewer 1 point 7. 

A column with mean differences and 95% confidence interval was added to Table 1.

12. Results, Lines 151-152. Please report comparisons for ALL variables collected, not just the age, sex and C3d/C3 (e.g. smoking status, BMI, etc.). 

Please see our previous answer to reviewer 1 point 3 (reason 1).

Unfortunately, we cannot report additional comparisons between cases and controls. We did not collect BMI and smoking variables for our control cohort. The other values in Table 2 are related to STGD1 progression and it makes no sense to measure and report these for the control cohort. 

13. Results, Lines 153-157. These dot plots are very helpful, thank you. Please superimpose these dots with a box plot to illustrate also the first and third quartiles. 

We superimposed the dot-plots with a box plot. 

Figure 1 was changed.

14. Results, Lines 168-181. This paragraph confirms that the cases may indeed come from a qualitatively different "population' than the controls. As noted above, it is extremely important to describe how cases and controls were sampled, from where, etc.

Please see our previous answers to reviewer 1 point 1 and point 2. And please also review the answer to reviewer 2 point 2. 

15. Discussion, Line 196. Please clarify that by "disease progression" this manuscript refers to "rate of progression". 

Text was changed accordingly. 

Discussion, line 196: ‘Moreover, systemic complement levels were not associated with disease parameters or disease progression’ was changed to Moreover, systemic complement levels were not associated with disease parameters or the rate of disease progression.

16. Discussion. Please include a limitations section here. Please state at least the three most important limitations and how robust this study may be to these limitations. 

We added a paragraph to the discussion section that describes the limitations of our study. 

Discussion, line 256, limitations paragraph added:

Our study has several limitations. First, we present a cohort that was sufficiently sized to detect effect sizes as large as those previously detected in AMD patients with GA. However, this cohort is too small to detect lower effect sizes. Larger studies are needed to exclude the role of systemic complement activation in STGD1. Second, the inclusion of patients with a possible underlying inflammatory condition creates outliers which causes an extreme data bias. However, when omitting these patients, our results remained the same. Third, complement levels are known to be influenced by other factors including age, sex, BMI and smoking. [15, 32] Even though cases and controls were matched on the basis of age and sex, there may be residual confounding and unfortunately, we did not collect BMI and smoking variables for our control cohort. Therefore, it is not possible to control for these covariates when analyzing our data. This study is the first to investigate systemic complement activation in STGD1 and its potential association with disease progression, and should therefore be considered a pilot study. Ideally, complement activity should be monitored in individual patients as their disease progresses to establish the effect of the covariates on complement levels and to establish the effect of complement levels on disease progression.

Journal Requirements 

We checked our work with the PLOS ONE’s style requirements. 

2. Thank you for including your ethics statement: "Ethical approval was obtained by the local institutional review board (project number: 2017-3535). All participants provided written informed consent before inclusion in the study. ". 

Please amend your current ethics statement to include the full name of the ethics committee/institutional review board(s) that approved your specific study. Once you have amended this/these statement(s) in the Methods section of the manuscript, please add the same text to the “Ethics Statement” field of the submission form (via “Edit Submission”). 

Ethics statement was adjusted accordingly. 

Methods, Participants, line 92: ‘Ethical approval ..was board (project number: 2017-3535)’ was changed to Ethical approval was obtained by the CMO Arnhem-Nijmegen and the Radboud University Medical Center local institutional review board (project number: 2017-3535) 

Other adjustments 

1. Accidently swapped values in manuscript: When recalculating the mean difference, we noticed that p-values were swapped for the correlation analysis of sex with the complement levels. There was a significant difference in C3 between males and females (and not in C3d/C3 ratio as currently written in the manuscript). 

Table 2 was changed accordingly.

Abstract, Results, lines 19-21: ‘Within the patient cohort … smoking status’ was changed to Within the patient cohort, C3 levels were associated with sex (mean difference -134, p=0.001, independent samples t-test) and BMI (correlation coefficient 0.463, p<0.001, Spearman’s Correlation).

Results, lines 187-188: ‘Complement levels … (p=0.001, independent samples t-test).’ was changed to C3 levels were associated with sex and BMI in STGD1 patients. Female patients had a higher C3 ratio compared to males (p=0.001, independent samples t-test).

2. Reference was added: We added the recently published manuscript (PMID 33849814) as reference 46 

Reference 46. Dhooge PPA, Mulders TWF, Hoyng CCB. Commentary on “Evidence of complement dysregulation in outer retina of Stargardt disease donor eyes”. Redox Biology. 2021:101957. was added (and reference 45 was changed to 46).

---

## [Decision Letter · Decision Letter 1]

11 Jun 2021

Systemic complement activation levels in Stargardt disease

PONE-D-21-08972R1

Dear Dr. den Hollander,

We’re pleased to inform you that your manuscript has been judged scientifically suitable for publication and will be formally accepted for publication once it meets all outstanding technical requirements.

Kind regards,

Demetrios G. Vavvas

Academic Editor

PLOS ONE

Additional Editor Comments (optional):

Reviewers' comments:

Reviewer's Responses to Questions

**Comments to the Author**

1. If the authors have adequately addressed your comments raised in a previous round of review and you feel that this manuscript is now acceptable for publication, you may indicate that here to bypass the “Comments to the Author” section, enter your conflict of interest statement in the “Confidential to Editor” section, and submit your "Accept" recommendation.

Reviewer #1: All comments have been addressed

2. Is the manuscript technically sound, and do the data support the conclusions?

Reviewer #1: Yes

3. Has the statistical analysis been performed appropriately and rigorously? 

Reviewer #1: Yes

4. Have the authors made all data underlying the findings in their manuscript fully available?

Reviewer #1: Yes

5. Is the manuscript presented in an intelligible fashion and written in standard English?

Reviewer #1: Yes

6. Review Comments to the Author

Reviewer #1: The authors have satisfactorily addressed all of my comments. I congratulate them on this excellent work!

7. PLOS authors have the option to publish the peer review history of their article (what does this mean?). If published, this will include your full peer review and any attached files.

Reviewer #1: No

---

## [Editor Report · Acceptance letter]

18 Jun 2021

PONE-D-21-08972R1 

Systemic complement activation levels in Stargardt disease 

Dear Dr. den Hollander:

I'm pleased to inform you that your manuscript has been deemed suitable for publication in PLOS ONE. Congratulations! Your manuscript is now with our production department. 

Kind regards, 

on behalf of

Dr. Demetrios G. Vavvas 

Academic Editor

PLOS ONE